# Up to 100% Replacement of Natural Materials from Residues: Recycling Blast Furnace Slag and Fly Ash as Self-Leveling Cementitious Building Materials

**DOI:** 10.3390/ma16093350

**Published:** 2023-04-25

**Authors:** David Torréns-Martín, Lucía J. Fernández-Carrasco, María Teresa Blanco-Varela

**Affiliations:** 1Department of Civil and Environmental Engineering (DECA), Universitat Politècnica de Catalunya (UPC), 08034 Barcelona, Spain; 2Instituto de Ciencias de la Construcción Eduardo Torroja (IETcc-CSIC), C/Serrano Galvache 4, 28033 Madrid, Spain

**Keywords:** wastes, recycle, slag, fly ash, calcium sulfate, microstructure, mechanical properties

## Abstract

The objective of this research is to study the use in the construction industry of recycled slag (SL) and fly ash (FA) using from 0.1 to 5% calcium sulfate (wC$\bar{S}$). These wastes have been used to make ternary mixture systems and evaluated in terms of technological properties as cementitious materials for building applications. Studying their micro-structure as hydration products, setting times and mechanical properties shows a way to develop new mixtures from high proportion of waste, which are alternatives to the traditional nature ternary systems: Portland cement (PC), calcium aluminate cement (CAC) and calcium sulphate (C$\bar{S}$). Based on previous work with natural products, the selected SL/FA ratios were 9 and 2.3 and the sulphate contents were 0, 1 and 5%. The water/binder ratio used for these cementitious mixes was 0.4. The specimens prepared for strength determination were prisms of 10 × 10 × 60 mm. The pastes were prepared and cured at 20 °C and 98% relative humidity for 1 day and then either stored at 20 °C at 98% humidity (dry) or immersed in distilled water (wet) for 14 and 28 days. The results showed that both FA and SL mixed with C$\bar{S}$ produce ettringite after 28 days of setting, and this phase was the main crystalline hydrated product in all mixes. Calcium sulphate stimulates the hydration reactions of the mixes and the strength increases when the C$\bar{S}$ content is higher due to ettringite formation, while the setting time decreases, as happened in the systems prepared with natural materials.

## 1. Introduction

The ternary systems of Portland cement, calcium aluminate cement and gypsum are complex, they are widely used in a variety of repair projects, and can be used in self-levelling flooring due to their expansion/shrinkage properties [1,2].

Analysis of numerous publications of the twenty-first century shows that the cement research community has reached an unequivocal consensus on the need to take greater care of the environment and to seek solutions to the problem of the carbon footprint of cement production [3,4,5]. According to UN projections, the world’s population will reach 10.4 billion by 2100 [6]. There are two important implications associated with this significant population growth: the need for housing for this population and the need to build in a sustainable manner to avoid negative impacts on climate change. In fact, it is a necessity to reuse the resources generated by the excessive and unsustainable consumption. Discarded materials are now accumulating in landfills without any use and contributing to soil and water pollution through their leachate. From a sustainability perspective, the use of by-products in the cement and concrete industry has significant environmental benefits. These materials are used not only to reduce the energy consumption and CO_2_ emissions of the cement manufacturing process, but also for their well-known durability properties [5]. 

Some construction solutions require fast installation processes, and this motivates the search for new combinations of cementitious materials that can provide good properties in a shorter curing period. Among other materials are those formed within the Portland cement, calcium aluminate cement and calcium sulphate (PC/CAC/C$\bar{S}$) systems, which have a rapid setting and strength development [1,2]. However, depending on the C$\bar{S}$ content, some durability problems may occur due to an expansion phenomenon, which limits their application in the construction industry [7,8].

Granulated blast-furnace slag (GBFS) is defined as a non-metallic product comprising silicates and aluminosilicates that are formed under liquid conditions together with the iron in the blast furnace. GBFS has been successfully used in the construction and transport industries as an aggregate for concrete, as a constituent of cement or in the production of thermal insulation [9,10]. Fly ash is produced in power plants by the combustion of coal. Fly ash can be of two main types: silico-aluminous and silico-calcareous. Both are considered supplementary cementitious materials (SCMs) as they have pozzolanic properties which, when added, can have hydraulic properties [11,12]. Different systems formed by SCM have been reported in the literature. Hwang and Shen [13] investigated the influence of slag and fly ash on the hydration of Portland cement (PC). They found a delay in hydration as the slag and/or fly ash content increased. Several studies [14,15] show that the strength development is greater when the slag content is increased and while the results are poor when the fly ash content is increased. Work has been done to model the behavior of these systems [16,17]. The hydration mechanism is divided into three processes; an initial inactive period, a second acceleration period and a final diffusion period. The models have the calcium hydroxide content as a limiting parameter for the SCM reactions.

CAC and slag mixtures [18,19,20,21] yield the typical hydration products of CAC, CAH_10_, C_2_AH_8_ and AH_3_. The proportions of each depend on the hydration temperature. Silicates from C_2_ASH_8_ disappear in the conversion reactions to form C_3_AH_6_ giving high strength. As the temperature increases, the presence of C_2_ASH_8_ decreases, C_3_AH_6_ formation increases and the strength development decreases. CAC and fly ash binders show a decrease in strength development as the fly ash content increases. CAC conversion is not avoided and C_2_ASH_8_ is formed at longer ages [22]. When calcium sulphate is added to these binders, the hydration products are different. Hexagonal hydrates are not formed and ettringite is the main hydrated phase. As the fly ash content increases, the ettringite formed is greater [23]. The system formed from slag and fly ash activated with NaOH shows as the main hydration product C-S-H with a high number of substitutions of Si by Al [24]. Strength development is better for blends containing 25–30% fly ash, but the activation of fly ash contributes more to longer ageing [25].

The addition of C$\bar{S}$ to PC and slag blends gives shorter setting times, showing that the slag does not participate in the setting time; small additions of C$\bar{S}$ improve the development of strength. These blends have a phenomenon of expansion which is higher when the amount of C$\bar{S}$ is greater. This fact shows that the expansions are due to ettringite formation [26,27,28]. Several studies [29,30,31,32,33] confirm the ettringite formation in mixtures of slag with C$\bar{S}$ where the expansion phenomenon is produced. In these studies, the slag was activated by different agents (lime, sulfoaluminate cements, potassium or sodium hydroxide).

The activation of compositions within the fly ash/PC system by means of C$\bar{S}$ was studied by Poon et al. [34] who found a high strength development at early ages when the activator was anhydrite, which was less effective in increasing strength at the later ages strengths than gypsum; this high strength was attributed to ettringite. In contrast, Sivapullaiah and Moghal [35] in studies investigating the C$\bar{S}$ influence in fly ash with lime mixtures see an indicated increase in strength when gypsum is added. This increase was attributed to the formation of a silicate-calcium aluminate-sodium hydrate along with the hydrated calcium silicate.

The aim of this work is to study the systems formed by slag, fly ash and C$\bar{S}$ and to determine their use as construction materials. Their microstructure, hydration products and mechanical properties were studied and compared with the ternary systems formed by PC/CAC/C$\bar{S}$ from natural materials in order to establish a link between them. Several blends were studied by X-ray diffraction (XRD), Fourier transform infrared spectroscopy (FTIR), scanning electron microscopy (SEM/EDX) and various mechanical tests to achieve the objective.

## 2. Materials and Methodology

### 2.1. Materials

The materials used were granulated blast furnace slag, fly ash and sulphates. The granulated blast furnace slag (SL) had a mean particle size of 15 µm, determined by laser diffraction, and a very small proportion of particles larger than 100 µm. Figure 1 shows the distribution function of the particle size of the waste materials. The fly ash (FA) presented a more heterogeneous particle size composition—they were between 25 and 300 µm. Regarding the sulphates, the average particle size composition was in the range of 10–40 µm, but also a lower maximum size of 160 µm was observed (see Figure 1). 

The chemical composition of the wastes was determined using a PHILIPS PW 2400 X-ray fluorescence spectrometer with a PW 2540 VTC sample changer, both from Philips, Amsterdam, The Netherlands. Table 1 shows the chemical composition of the slag, fly ash and calcium sulphate supplied by Algiss-Uralita. According to the analysis results, the fly ash is type F in the ASTM classification. 

### 2.2. Methods

For this study, ten SL/FA/C$\bar{S}$ formulations were prepared in a mixer by mixing the powdered waste raw materials for 1 h at 90 rpm. Figure 2 shows the compositions studied for this ternary system—according to another previous study [36].

The SL/FA used was 9 and 2.3 and the calcium sulphate content was 0, 1 and 5%—six binary and four ternary compositions. To prepare the pastes, a water/solid ratio of 0.4 (in order to compare with previous research) was used and the pastes were cured in two different environments: (1) in a curing chamber with 98% relative humidity and 20 °C temperature, and (2) in the same chamber but curing under distilled/deionized water.

To stop the hydration process and remove the unreacted water, the pastes hydrated for 14 and 28 days were ground with acetone in a porcelain mortar and then filtered [37]. The powder was washed on the filter with additional amounts of acetone and ethyl ether and finally dried in air in a desiccator.

#### 2.2.1. Chemical and Mineralogical Characterization

The mineralogical composition of the samples was determined by X-ray diffraction (XRD) using a Siemens D500 instrument, Munich, Germany and an FTIR-Boehm MB-120 Fourier transform infrared spectrophotometer with a frequency range of 450 to 4000 cm^−1^. A JEOL JSM-6300, New Delhi, India, Scanning Electron Microscope (SEM) and an attached LINK ISIS-200 Energy Dispersive X-ray Analysis (EDX) were used to obtain more detailed information on the morphology and elemental composition of the samples.

#### 2.2.2. Mechanical Strength Measurements 

Specimens of 1 × 1 × 6 cm were prepared according to the Koch–Steinegger method [38] and cured in a chamber at 20 °C and 95% relative humidity (RH). The specimens were demoulded after 24 h. Half of the specimens were cured at 20 °C and 95% RH and the other half were maintained at 20 °C under distilled/deionized water. Mechanical tests were performed after 14 and 28 days of hydration. 

The setting time measurements were carried out in accordance with standard UNE-EN 196-3:2005 + A1:2009 [39], but with a change in the mass, using 100 g of sample. The measurements were carried out using an IBERTEST, Barcelona, Spain model IB-32-056 E instrument.

## 3. Results and Discussion

### 3.1. Mineralogical Characterization

The raw materials were characterized by XRD and FTIR (Figure 3). The XRD pattern of SL shows a halo between 25–35° 2θ, characteristic of amorphous glassy materials; there are two low intensity reflection lines assigned to merwinite (Ca_3_Mg-Si_2_O_8_). The FTIR spectrum of SL shows a main broad band at 1050–850 cm^−1^, which we assigned to ν_3_ Si-O vibrations, and a band at 520 cm^−1^ due to ν_4_ O-Si-O vibrations of the silicate tetrahedron. In the 600–800 cm^−1^ region a smaller band appears due to ν_3_ Al-O vibrations of the aluminate tetrahedron.

The XRD pattern of fly ash also shows an amorphous halo and several reflections due to the presence of quartz, mullite and hematite as crystalline phases. The FTIR spectrum shows the main ν_3_ Si-O band at a higher wave number (1086 cm^−1^) than the FTIR spectrum of slag, due to the presence of quartz and mullite and a glassy phase where silicates have a three-dimensional structure, whereas silicate in slag mainly forms dimers [40].

Characteristic reflections and bands of basanite (CaSO_4_·0.5H_2_O) can be seen in the XRD pattern and FTIR spectrum of calcium sulphate [41] (see Figure 3).

#### 3.1.1. Development of Hydrated Phases

The diffraction patterns (Figure 4) show the phase evolution when C$\bar{S}$ is added to wastes in binary compositions, with ambient and underwater curing. 

After 14 days of hydration of the FA/C$\bar{S}$ 5% sample, a small amount of ettringite was detected, which was higher in the immersed samples. In contrast, the mixture formed by slag and C$\bar{S}$ provides a major hydration. The diffraction patterns show a higher ettringite production at 28 days and the halo due to the glassy phases was less intense in all DRX diffraction patterns. In the FTIR spectra of the FA/C$\bar{S}$ samples (Figure 5), ettringite formation is visible, but amorphous phases are also detectable. The silicate phases have been consumed; the band at 520 cm^−1^ associated with Si-O-Si bending of anhydrous silicates [42] disappears and the vibrational band of Si-O is sharper, indicating the consumption of anhydrous silicate phases and the formation of phases such as hydrated calcium silicates.

In the ternary systems, mixtures formed by SL/FA/C$\bar{S}$ were presented as main hydrated phases of the ettringite. Figure 6 displays a semi quantitative analysis based on the ettringite diffraction line located at 9.08° 2θ; the binary systems formed by slag and fly ash were ignored because the detected ettringite level was low. The ettringite content was higher in the submerged samples. The binders with more initial C$\bar{S}$ incorporated have a higher ettringite production and the systems with a high proportion of fly ash (70/30) also have this proportion, mainly the compositions with only 1% C$\bar{S}$.

Both facts, higher ettringite production at higher C$\bar{S}$ and lower production for the systems with higher fly ash content, are due to waste activation. In the diffraction patterns after 28 days, the anhydrous phases of the fly ash are still visible, indicating that they have a slow reaction process. Singh et al. [29] found that dihydrated calcium sulphate can activate the slag when hydrated in the presence of Na or Fe. In the case of this study, C$\bar{S}$ was used as a hemihydrate and Fe and Na were also present in the system. These conditions allow for the activation of the slag, resulting in the formation of ettringite. In the case of fly ash, several studies [42,43] agree on the need for an agent that attacks the fly ash structure, breaking the Si-O-Si and Al-O-Si bonds and activating the waste. In this case, the C$\bar{S}$ cannot attack the fly ash by itself and would need the presence of an activator such as Ca(OH)_2_ which increases the pH. The ettringite formed in the fly ash/C$\bar{S}$ mix can be attributed to the free lime present in the fly ash. Typically, this waste contains 1% free lime [44], which can react with the C$\bar{S}$ to form ettringite.

The FTIR spectra confirm these results (Figure 7). The ettringite production is lower when the fly ash content is higher. Ettringite bands (1120 and 620 cm^−1^) are only visible in the composition with 5% C$\bar{S}$ in the 70/30 composition. At 971 and 532 cm^−1^ are the bands corresponding to Si-O stretching and Si-O-Si bending, indicating the presence of a gel such as C-S-H. For the 70/30 compositions the stretching band moves to 968 cm^−1^, being at higher values in the 90/10 compositions; this fact could be due to a lower polymerization of the gel for these mixtures. No difference was observed between the two types of hardening.

#### 3.1.2. Morphologies and Element Composition of the Developed Phases

The morphology development in the different mixtures was studied by SEM. The morphology of the phases in the 90/10 samples was poorly developed in both curing systems (Figure 8a). In the 1% C$\bar{S}$ composition, the developed morphology was like needles and was highly present in 5% C$\bar{S}$ systems (Figure 8b). EDX analysis shows that these crystals had the composition of ettringite (Figure 8c). No differences were associated with curing, as in both cases they had similar crystal morphologies with a size of approximately 8 μm in length.

The 70/30 compositions show a different differentiated morphology. There were spheres coated with a dense layer (Figure 9a). EDX analysis performed on this form of microstructure indicated that these spheres were unreacted fly ash and the layer was a gel (Figure 9-1 and Figure 9-2). In the absence of C$\bar{S}$ there were CaCO_3_ needles. This morphology suggests that carbonates are present in the form of aragonite [42]. The FTIR spectra (Figure 5) also show this form of carbonate. When C$\bar{S}$ was added, plate and needle crystals were formed (Figure 9b). The EDX analysis shows CaCO_3_ and ettringite, respectively (Figure 9-3). In the 5% C$\bar{S}$ system, the presence of needles was higher, indicating greater ettringite production. More ettringite in the form of needles was visible in the submerged compositions.

The EDX analysis carried out on the developed gel shows its elemental composition. Figure 10 shows the Al/Ca ratio in relation to the Si/Ca ratio. The Al/Ca ratio is higher than for a C-S-H gel, indicating a substitution of Si by Al. This incorporation of Al is higher in the 70/30 compositions and lower when the C$\bar{S}$ is higher. When the C$\bar{S}$ is added to the system, it reacts with the aluminates to form ettringite, preventing it from entering the gel.

### 3.2. Mechanical Strength Behavior and Setting Times

The strength development depends on the initial composition (Figure 11), being slightly greater in the samples immersed in water. The compositions containing only slag developed the greater strength. The presence of fly ash in the mixes decreases the strength, but in contrast, as the C$\bar{S}$ content increases, the strength is higher. This behavior is due to the fact that fly ash does not react and causes a decrease in strength. When more C$\bar{S}$ is added, ettringite production is greater, which increases the strength development.

### 3.3. Comparison of Results with PC/CAC/C$\bar{S}$ Systems from Natural Materials

Previous works [1,2,16] have already shown that the main hydration products in the ternary PC/CAC/C$\bar{S}$ systems are ettringite and C-S-H gel with Al incorporated in its structure. These hydration products are present in the studied systems, with a higher proportion of ettringite when the C$\bar{S}$ proportion is higher.

The mechanical behavior shows differences to the PC/CAC/C$\bar{S}$ systems. The studies [45,46] show that as the aluminate content increases, the setting time decreases and the strength development is greater. In the case of this study, the phases with high aluminates were fly ash, but an increase of this compound in the mixes leads to a lower strength and a longer setting time (Figure 12). This discrepancy is explained by the lack of activation of the fly ash or the unavailability of the aluminates to be reacted or incorporated in the C-A-S-H gels.

In CP/CAC/C$\bar{S}$, the microstructures depended on the CP/CAC and CAC/C$\bar{S}$ ratios [1,8,47,48]. For example, systems with CAC/C$\bar{S}$ ratios showed increased ettringite needles and microstructures typical of CAC [49]. Systems with high CP/CAC ratios have a gel C-S-H and ettringite in the form of small spheres [2]. Microstructures developed in the SL/FA/C$\bar{S}$ present crystallizations in the form of needles. These needles were carbonates for low C$\bar{S}$ fractions and ettringite when this fraction increased. There was also a gel formed by hydrated silicate-aluminate calcium.

## 4. Conclusions

A very high percentage of waste was used to replace natural products in the production of self-leveling mixes of the CAC-CP-CS type, thus contributing to the SDGs to mitigate climate change. The conclusions of the study were as follows.

The main hydrated products in these systems are ettringite and a type of C-S-H gel. The production of ettringite is determined by the reaction between the slag and the C$\bar{S}$. A gel similar to C-S-H is formed, but with substitution of Si by Al, which is a C-A-S-H gel.

The microstructures developed depend on the initial composition. Carbonates or ettringite appear in the form of needles depending on the proportion of C$\bar{S}$.

The increase in strength is greater when the C$\bar{S}$ content is greater and is due to ettringite formation, but the gel phases also contribute.

The fly ash is not activated by the calcium sulphate, which is explained by the low OH species present in the process, which prevents the Si-O bonds from breaking.

## Figures and Tables

**Figure 1 materials-16-03350-f001:**
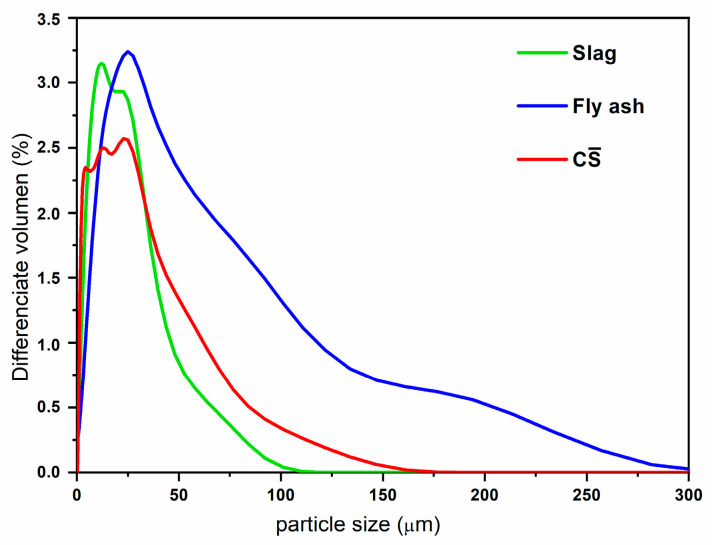
Particle size of raw materials obtained by laser granulometry.

**Figure 2 materials-16-03350-f002:**
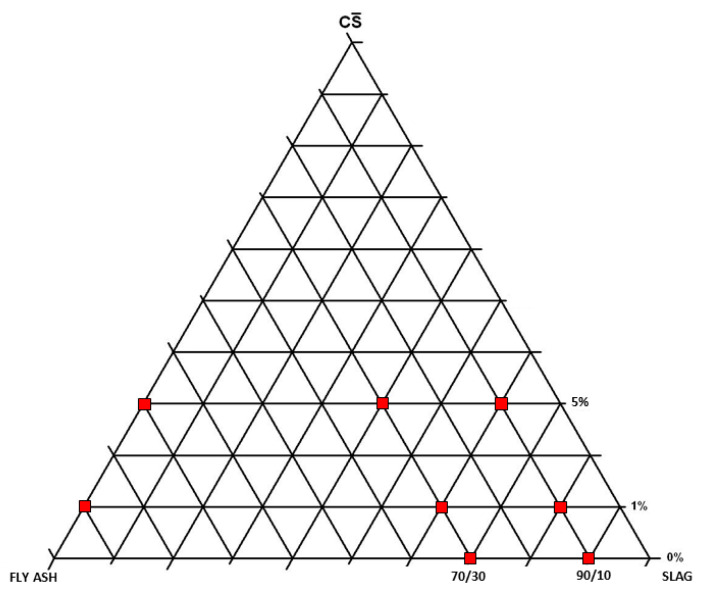
Selected compositions of the slag/fly ash/calcium sulfate system.

**Figure 3 materials-16-03350-f003:**
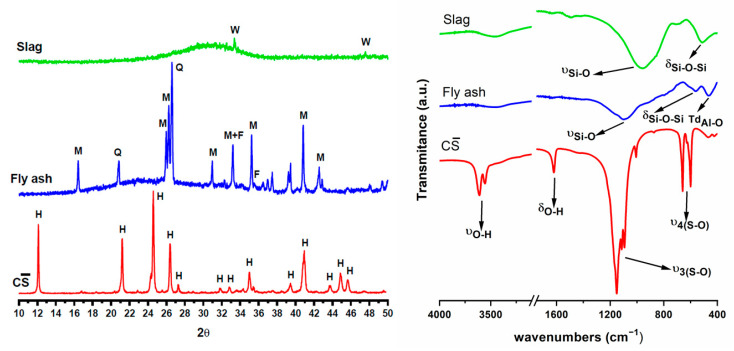
Characterization of raw materials. W: merwenite, M: mullite, Q: quartz, F: hematite, H: CaSO_4_·1/2H_2_O.

**Figure 4 materials-16-03350-f004:**
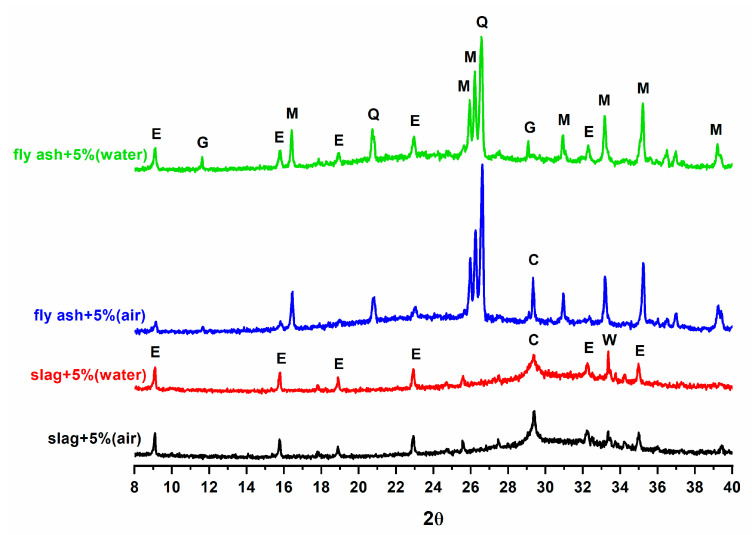
Diffraction patterns of wastes with 5% of C$\bar{S}$ at 28 days cured in two environments E: ettringite, G: gypsum, C: Calcite, W: merwenite, M: mullite, Q: quartz, H: CaSO_4_·1/2H_2_O.

**Figure 5 materials-16-03350-f005:**
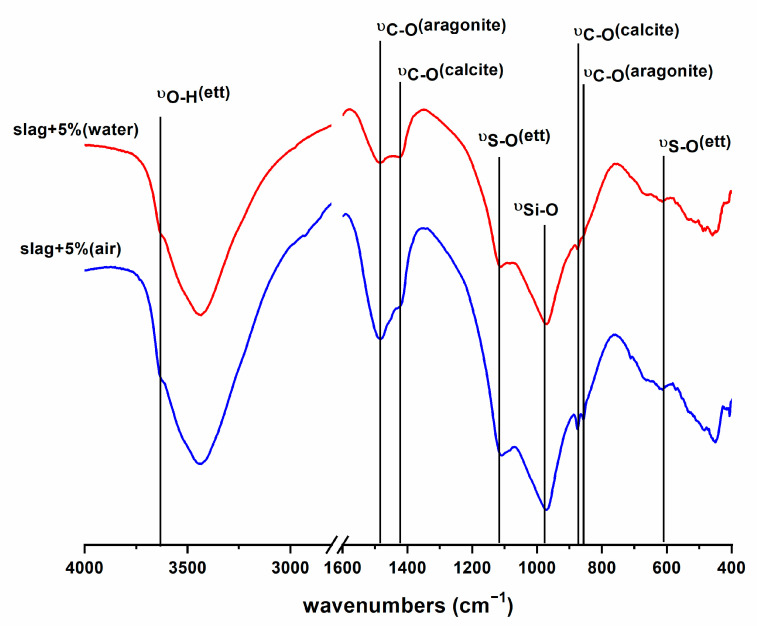
FTIR spectra of slag with 5% of C$\bar{S}$ at 28 days cured in two environments.

**Figure 6 materials-16-03350-f006:**
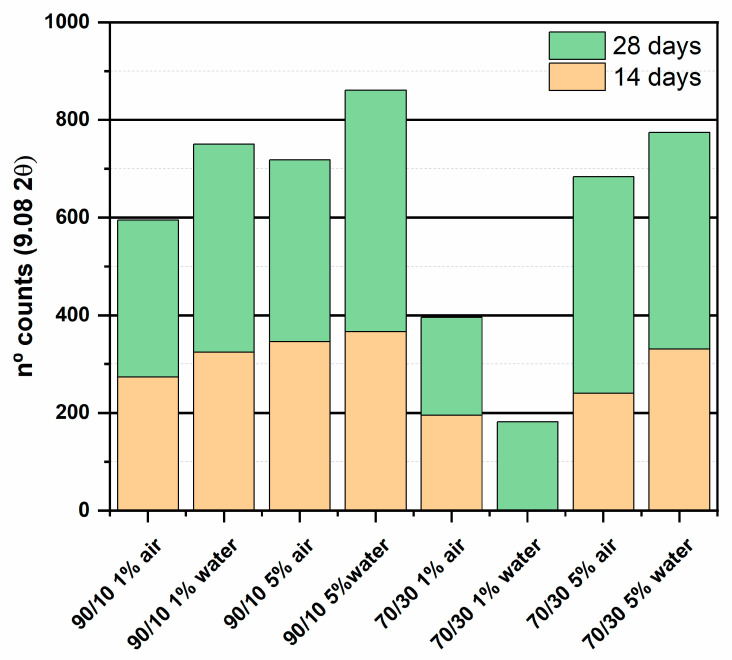
Numbers of counts of ettringite diffraction line at 9.08° 2θ.

**Figure 7 materials-16-03350-f007:**
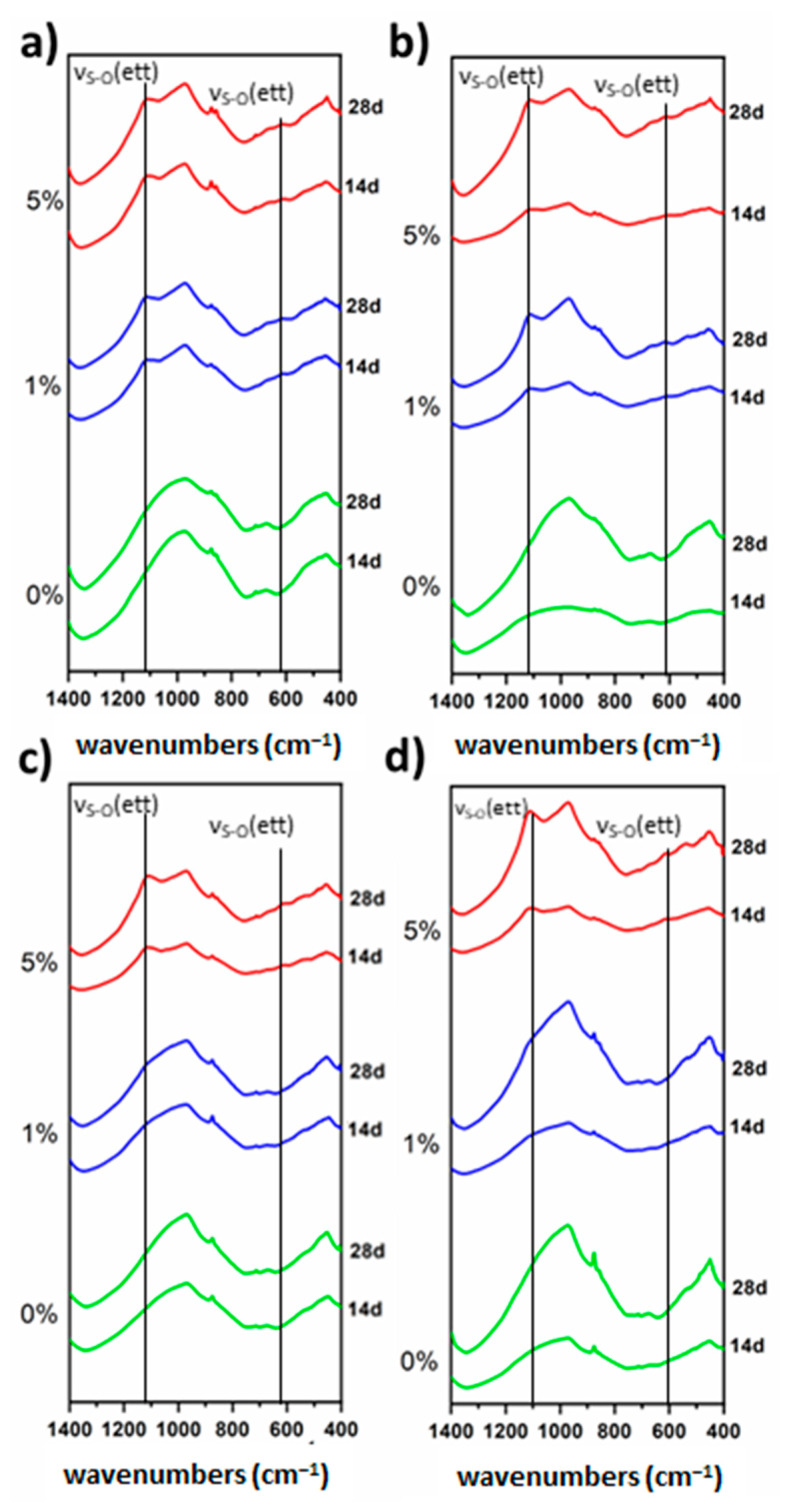
FTIR spectra of ternary system SL/FA/C$\bar{S}$. (**a**) 90/10 air, (**b**) 90/10 water, (**c**) 70/30 air, (**d**) 70/30 water.

**Figure 8 materials-16-03350-f008:**
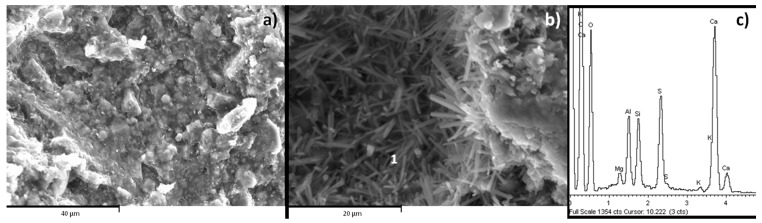
SEM images at 28 days for the 90/10 system. (**a**) 90/10 5% C$\bar{S}$ water, (**b**) 90/10 1% C$\bar{S}$ air, (**c**) EDX analysis point 1.

**Figure 9 materials-16-03350-f009:**
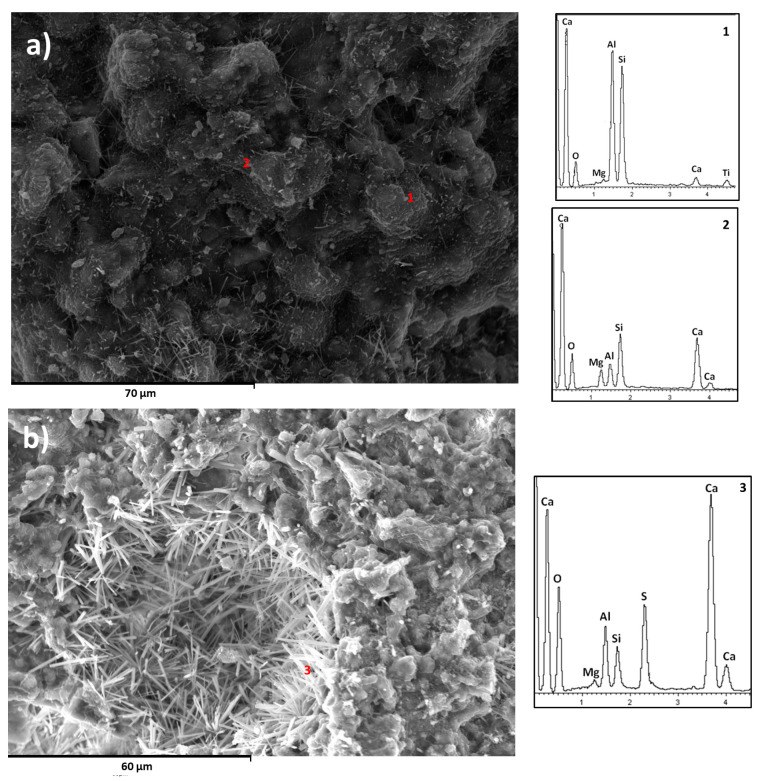
(**a**) SEM image at 28 days for the composition 70/30 0% C$\bar{S}$ in air. (**b**) SEM image at 28 days for the composition 70/30 5% C$\bar{S}$ under water. EDX analysis 1: sphere, 2: dense film, 3: needles.

**Figure 10 materials-16-03350-f010:**
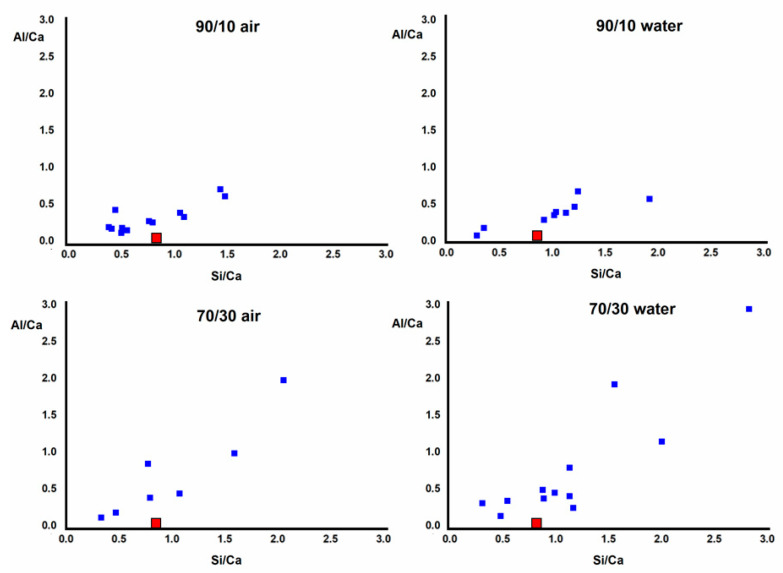
Al/Ca vs. Si/Ca ratio through EDX analysis made in the ternary system SL/FA/C$\bar{S}$. The red square corresponds to a C-S-H gel structure tobermorite ratio.

**Figure 11 materials-16-03350-f011:**
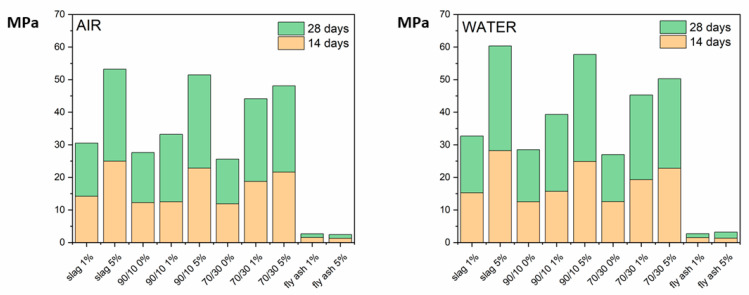
Compression strength measurements at 14 and 28 days.

**Figure 12 materials-16-03350-f012:**
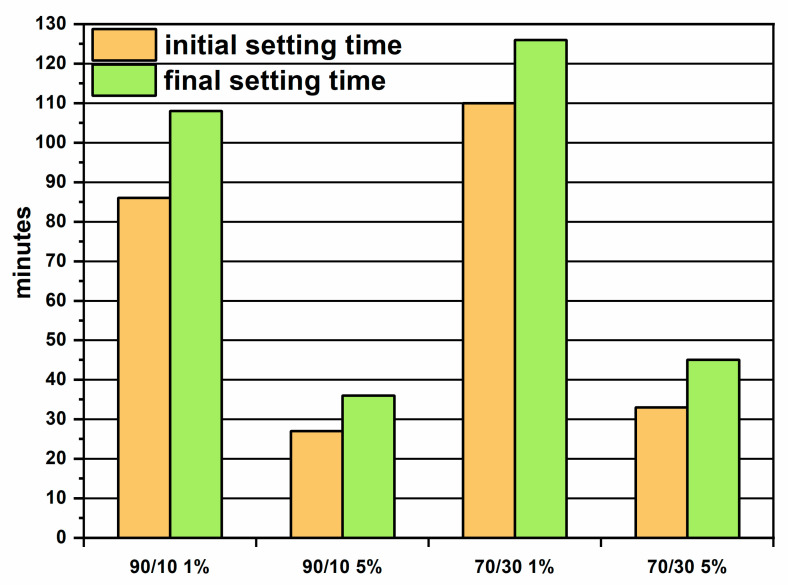
Setting time for ternary systems SL/FA/C$\bar{S}$.

**Table 1 materials-16-03350-t001:** Chemical compositions of residue materials (% of oxides).

	SiO_2_	Al_2_O_3_	CaO	Fe_2_O_3_	SO_3_	MgO	TiO_2_	MnO	K_2_O	SrO	P_2_O_5_	Na_2_O
**Slag**	35.01	11.35	41.27	0.51	-	8.68	0.53	0.22	0.39	-	0.04	<1
**Fly ash**	48.12	30.36	7.91	3.24	0.39	1.9	1.45	-	0.59	0.36	1.81	-

## Data Availability

Not applicable.

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
