# Peer review of "Up to 100% Replacement of Natural Materials from Residues: Recycling Blast Furnace Slag and Fly Ash as Self-Leveling Cementitious Building Materials"

_materials, 2023, doi:10.3390/ma16093350_

Round 1
Reviewer 1 Report
Reviewer comment: This paper deals with the use of synergic industrial waste mixtures instead of cement as cementing construction materials; The compositions in the slag (SL), fly ash (FA) and up to 5% of calcium sulfate (CSÌ…) ternary systems (100% of residues) were evaluated in terms of their mineralogical and technological properties as building applications. However, some major comments are listed as follows:
Comments and suggestions:
1- In the abstract section, the author should explain “The selected SL/FA ratios were 9 and 2.3, and the proportion of sulfates were 0, 1 and 5%.”
2- Generally, in the introduction section, the author should be listed along with the references that you used in the literature review, such as (page 2, lines 30 to 36).
3- Page 2 Line 49, modify "Blast furnace slags (GBFS) " to “Granulated blast furnace slag (GBFS)”.
4- The quality of the all figures 1and 2 is weak
5- Check table number Page 3 at line 100.
6- Please mention the specification for “In order to stop the hydration process and to remove the non-reacted water, the pastes hydrated for 14 and 28 days were grounded in a porcelain mortar with acetone and then filtered.” at page 4 line 139
7- The author should explain the meaning of v3, and v4 in the line 161 at page 5.
8- Figure 10 is not clear to read the data.
9- Section 3.2, please mention the size of tested samples, and attach the experimental photo.
10- Page 11, line 286, please explain more.
11- Figure 12: Modify the start and end setting times to “initial and final setting times”.
12- All references must be checked.
13- In general, please update some references and citations with recent, relevant, and internationally indexed (2022-2021-2020) literature.
14- The following references can be cited for the analysis of microstructure and microscopic mechanism.
· https://doi.org/10.1016/j.conbuildmat.2022.129300
· https://doi.org/10.1007/s41062-021-00657-z
· https://doi.org/10.1016/j.conbuildmat.2021.124491
15- Please avoid reference overkill/run-on, i.e. do not use more than 3 references per sentence.
Author Response
Reviewer comment: This paper deals with the use of synergic industrial waste mixtures instead of cement as cementing construction materials; The compositions in the slag (SL), fly ash (FA) and up to 5% of calcium sulfate (CSÌ…) ternary systems (100% of residues) were evaluated in terms of their mineralogical and technological properties as building applications. However, some major comments are listed as follows:
Comments and suggestions:
- In the abstract section, the author should explain “The selected SL/FA ratios were 9 and 2.3, and the proportion of sulfates were 0, 1 and 5%
It has been explained.
- Generally, in the introduction section, the author should be listed along with the references that you used in the literature review, such as (page 2, lines 30 to 36).
References has been added.
- Page 2 Line 49, modify "Blast furnace slags (GBFS)" to “Granulated blast furnace slag (GBFS)”.
This modification has been implemented.
- 4-The quality of the all figures 1and 2 is weak
The quality of Figures 1 and 2 are improved
- Check table number Page 3 at line 100.
That was also checked.
- Please mention the specification for “In order to stop the hydration process and to remove the non-reacted water, the pastes hydrated for 14 and 28 days were grounded in a porcelain mortar with acetone and then filtered.” at page 4 line 139
A reference was added:
R.L. Day. Reactions between methanol and portland cement paste. Cement and Concrete Research Volume 11, Issue 3, May 1981, Pages 341-349
- The author should explain the meaning of v3, and v4 in the line 161 at page 5.
This is the vibrational modes of the molecules: some basic of Infrared spectra. I will not be mention that in the manuscript.
- Figure 10 is not clear to read the data.
The Figure 10 has been updated.
- 9-Section 3.2, please mention the size of tested samples, and attach the experimental photo.
The size of tested samples is already mentioned in the materials and methods part, but a picture has been updated.
- Page 11, line 286, please explain more.
Not clear what you ask: page 11 is Figure 1 and line 286 correspond to references.
- Figure 12: Modify the start and end setting times to “initial and final setting times”.
Done
- All references must be checked.
The references have been checked.
- In general, please update some references and citations with recent, relevant, and internationally indexed (2022-2021-2020) literature.
- The following references can be cited for the analysis of microstructure and microscopic mechanism.
- https://doi.org/10.1016/j.conbuildmat.2022.129300
Thermal, mechanical and microstructural properties of sustainable concrete incorporating Phase change materials
Mostafa M. Alsaadawi a b, Mohamed Amin c, Ahmed M. Tahwia a
- https://doi.org/10.1007/s41062-021-00657-z
Synergistic influence of metakaolin and slag cement on the properties of self-compacting fiber-reinforced concrete
Amal Raia, Ahmed Tahwia, Ahmed Hassanin Abdel Raheem & Mohamed Abd Elrahman
- https://doi.org/10.1016/j.conbuildmat.2021.124491
Durability and microstructure of eco-efficient ultra-high-performance concrete
Ahmed M. Tahwia a, Gamal M. Elgendy b, Mohamed Amin c
I do not see the relation of these papers.
15- Please avoid reference overkill/run-on, i.e. do not use more than 3 references per sentence.
Done

Reviewer 2 Report
In this paper, the authors discuss the extraction of raw materials from nature and the production of cement results in high CO2 emissions to the atmosphere and represent significant energy consumption. In this manuscript, the use of synergistic mixtures of industrial wastes instead of cement as construction materials for cementation is proposed; compositions in ternary systems of slag (SL), fly ash (FA) and up to 5% calcium sulfate (CSÌ…) (100% residues) were evaluated in terms of their mineralogical and technological properties as construction applications.
The study of their microstructure, hydration products, setting times and mechanical properties shows a path for the development of new mixtures with a high percentage of waste, alternatives to traditional ternary systems: Portland cement (PC), calcium aluminate cement (CAC) and calcium sulfate (CSÌ…). The selected SL/FA ratios were 9 and 2.3, while the sulfate percentage was 0, 1, and 5 percent.
The outcomes showed that both FA and SL combined with CSÌ… produced mainly ettringite in all the compositions studied but also a kind of CASH gel. Furthermore, calcium sulfate stimulates the hydration responses of the mixtures, and the power grows when the proportion of CSÌ… is higher due to the formation of ettringite while the setting time decreases. In conclusion, using adequate residuals is a route to obtain cemetery materials without incorporating natural fabrics that play an essential role in sustainable consumption and display processes and perform well for use as construction or construction systems.
The authors propose a very interesting background. The work is surrounded by a good proposal. The paper is readable and fluent. In addition, the tables and proposals are in line with the purpose of the paper. The paper sounds good scientifically.
Author Response
Thank you so much for your comments on the paper

Reviewer 3 Report
General Comments:
  This study deals with the systems formed by slag, fly ash and CSÌ… – all residues- and determinate their use as building materials. In order to achieve this objective, authors carried out the observation of their microstructure, hydration products and mechanical properties and comparing with the ternary systems formed by Portland cement/calcium aluminate cement/calcium sulfate from nature materials. Furthermore, authors carried out the study of the several mixtures with the help of X-ray diffraction (XRD), infrared spectroscopy (FTIR), electron microscope (SEM/EDX) and different mechanical tests. Especially it is considered that the originality of this paper is a very high percentage of wastes replaceing natural products in the production of self-leveling mixes of the CAC-CP-CS type to contribute to the SDGs to help mitigate climate change. It is considered that these results in this paper is very useful information for this area of this research field in the future.
However, it is considered that there are several points which reviewer cannot understand the meaning of sentence in this paper. Therefore, this paper is required to be improved enough to be published in the Journal of Materials.
Specific Comments:
1) At line 1 at the page of 1:
Reviewer cannot understand a meaning of "self-leveling cement" of the title of paper. A little plain title. Should make “leveling” a different word if possible.
2) Figures on the paper:
All figures are indistinct. Should change them more clearly
3) At line 99 to 154 at the page of 3 to 5:
There is no explanation of the calcium sulfate (CSÌ…) as the industrial by-product really.
In the same way as blast furnace slags and fly ash, please explain calcium sulfate.
4) At line 134 to 135 at the page of 4:
Even if water/solid ratio = 0.4 changes, does this knowledge not change?
Should explain the reason of decision of water/solid ratio = 0.4.
Should describe the mix proportion of all samples in this paper.
5) At line 151 at the page of 4:
There is not information on the mechanical tests. Should describe the mechanical tests in this paper.
6) At line 153 at the page of 5:
The word of “g” is enough for the unit of “gr”.
Author Response
Specific Comments:
1) At line 1 at the page of 1:
Reviewer cannot understand a meaning of "self-leveling cement" of the title of paper. A little plain title. Should make “leveling” a different word if possible.
There are a lot of specific self-leveling cementing materials within the formulation of CAC/PC/CSÌ… and it is well known by the specialized community. It is not possible to modify that because if not this waste cementing systems are not going to be associated with the correct application.
2) Figures on the paper:
All figures are indistinct. Should change them more clearly
We have updated that.
3) At line 99 to 154 at the page of 3 to 5:
There is no explanation of the calcium sulfate (CSÌ…) as the industrial by-product really.
In the same way as blast furnace slags and fly ash, please explain calcium sulfate.
The Calcium sulfate has been provided by Algiss-Uralita and is a waste product from plasterboard; this information has been added in the text.
4) At line 134 to 135 at the page of 4:
Even if water/solid ratio = 0.4 changes, does this knowledge not change?
Should explain the reason of decision of water/solid ratio = 0.4.
We made previous research with nature materials with this W/C ratio and for comparative proposes we maintain the same proportion.
Should describe the mix proportion of all samples in this paper.
It is the same in all samples.
5) At line 151 at the page of 4:
There is not information on the mechanical tests. Should describe the mechanical tests in this paper.
A reference has been added of the method applied.
6) At line 153 at the page of 5:
The word of “g” is enough for the unit of “gr”.
It has been modified.

Reviewer 4 Report
The manuscript “Up to 100% replacement of natural materials with waste: recycling of blast furnace slag and fly ash as self-leveling cement” describes the ternary system slag, fly ash and calcium sulfate from the view of hydration products, microstructure and mechanical properties. The recycling, saving resources and reuse of waste is important theme in nowadays.
The scientific soundness and the quality of presentation of the manuscript is low. In my opinion the manuscript has not the sufficient quality for publishing in Materials for these reasons:
· Novelty – the authors do not specify what is the novelty of their work. The hydration of fly ash and blast furnace slag have been described very detailed yet as well as their combination. The formation of ettringite is already known for many years. The results in manuscript do not bring any new findings.
· In the title, the authors specify, that developed mixtures will be used as self-leveling cement. In the whole manuscript, there is only one mention about self-leveling concrete. I miss the information about rheological properties, which play key role in self-leveling concrete.
· The section 1 (introduction) is not consistent. The information in each paragraph are not linked together. There should more about self-leveling cement. The part about blended systems BFS/fly ash/calcium sulfate should be extended and try to find actual references (the newest in the manuscript is from the year 2013).
· Low quality of pictures. Figure 2 and 11 are not legible. No consistency in color or black/white figures (fig. 3 x fig. 4).
· The section 3.1. should be in the section 2
· Different abbreviation for slag - SL x GBFS
· Line 123 – Algiss-Uralita is the supplier only of calcium sulphate or of slag and fly ash too?
· Tab. 1 – miss the chemical composition of calcium sulfate
· Fig. 2 – the summary of the prepared mixtures should be in the clear table
· Section 3 – there is only the description of the results, no scientific discussion
· In Fig.4 four different compositions are compared, in the Fig. 5 only two composition are compared. Why?
· The section 3.1.2. is confusing.
· Fig. 12 – why only 4 mixture were use for analysis of setting time? How these mixture were chosen?
· Poor English with typing errors (e.g. line 157 use y instead of and)
Author Response
In the title, the authors specify, that developed mixtures will be used as self-leveling cement. In the whole manuscript, there is only one mention about self-leveling concrete. I miss the information about rheological properties, which play key role in self-leveling concrete.
We detail the setting measurements of some compositions that is a good information for that.
- The section 1 (introduction) is not consistent. The information in each paragraph are not linked together. There should more about self-leveling cement. The part about blended systems BFS/fly ash/calcium sulfate should be extended and try to find actual references (the newest in the manuscript is from the year 2013).
There are references about FS and SL but not concerning mixing with gypsum. This is why there is not references about it.
- Low quality of pictures. Figure 2 and 11 are not legible. No consistency in color or black/white figures (fig. 3 x fig. 4).
That was improved
- The section 3.1. should be in the section 2
Not agree with that. 3.1 are results.
- Different abbreviation for slag - SL x GBFS
That was modify
- Line 123 – Algiss-Uralita is the supplier only of calcium sulphate or of slag and fly ash too?
Only the Calcium sulfate.
- Tab. 1 – miss the chemical composition of calcium sulfate
Because it is only calcium sulfate.
- Fig. 2 – the summary of the prepared mixtures should be in the clear table
This is in the Figure detailed and ratio mentioned in the test.
- Section 3 – there is only the description of the results, no scientific discussion
Not agree
- In Fig.4 four different compositions are compared, in the Fig. 5 only two composition are compared. Why?
These selected samples are the binaries compositions, more reacted samples as can be seen in XRD pattern are the slag/CS and only important information is extracted from their IR spectra.
- The section 3.1.2. is confusing.
We improve the text in order to clarify.
- Fig. 12 – why only 4 mixture were use for analysis of setting time? How these mixture were chosen?
Previous research on nature raw materials as reference.
- Poor English with typing errors (e.g. line 157 use y instead of and)
Improved

Round 2
Reviewer 1 Report
The authors made all the requested corrections.
Reviewer 4 Report
I am not able to evaluate this manuscript. In the corrected version the figures are covered with text and whole manuscript is confusing for me. Moreover there is no reaction of authors about my mention about the novelty of the manuscript.
Round 3
Reviewer 4 Report
The mauscript has been improved, but it has still low scientidfic quality. It is on the edge to be published or be rejected.